# Arsenic Removal via the Biomineralization of Iron-Oxidizing Bacteria *Pseudarthrobacter* sp. Fe7

**DOI:** 10.3390/microorganisms11122860

**Published:** 2023-11-26

**Authors:** Xia Fan, Hanxiao Zhang, Qian Peng, Yongliang Zheng, Kaixiang Shi, Xian Xia

**Affiliations:** 1College of Biology and Agriculture Resources, Huanggang Normal University, Huanggang 438000, China; fanxia@hgnu.edu.cn (X.F.); 15038881601@163.com (H.Z.); m18040651471@163.com (Q.P.); zylgolden@126.com (Y.Z.); 2National Key Laboratory of Agricultural Microbiology, College of Life Science and Technology, Huazhong Agricultural University, Wuhan 430070, China; 3Hubei Key Laboratory of Edible Wild Plants Conservation & Utilization, Hubei Engineering Research Center of Characteristic Wild Vegetable Breeding and Comprehensive Utilization Technology, College of Life Science, Hubei Normal University, Huangshi 435002, China

**Keywords:** arsenic removal, Fe(II)-oxidizing bacteria, iron mineral precipitates, biomineralization, *Pseudarthrobacter*

## Abstract

Arsenic (As) is a highly toxic metalloid, and its widespread contamination of water is a serious threat to human health. This study explored As removal using Fe(II)-oxidizing bacteria. The strain Fe7 isolated from iron mine soil was classified as the genus *Pseudarthrobacter* based on 16S rRNA gene sequence similarities and phylogenetic analyses. The strain Fe7 was identified as a strain of Gram-positive, rod-shaped, aerobic bacteria that can oxidize Fe(II) and produce iron mineral precipitates. X-ray diffraction, X-ray photoelectron spectroscopy, and energy-dispersive X-ray spectroscopy patterns showed that the iron mineral precipitates with poor crystallinity consisted of Fe(III) and numerous biological impurities. In the co-cultivation of the strain Fe7 with arsenite (As(III)), 100% of the total Fe and 99.9% of the total As were removed after 72 h. During the co-cultivation of the strain Fe7 with arsenate (As(V)), 98.4% of the total Fe and 96.9% of the total As were removed after 72 h. Additionally, the iron precipitates produced by the strain Fe7 removed 100% of the total As after 3 h in both the As(III) and As(V) pollution systems. Furthermore, enzyme activity experiments revealed that the strain Fe7 oxidized Fe(II) by producing extracellular enzymes. When 2% (*v*/*v*) extracellular enzyme liquid of the strain Fe7 was added to the As(III) or As(V) pollution system, the total As removal rates were 98.6% and 99.4%, respectively, after 2 h, which increased to 100% when 5% (*v*/*v*) and 10% (*v*/*v*) extracellular enzyme liquid of the strain Fe7 were, respectively, added to the As(III) and As(V) pollution systems. Therefore, iron biomineralized using a co-culture of the strain Fe7 and As, iron precipitates produced by the strain Fe7, and the extracellular enzymes of the strain Fe7 could remove As(III) and As(V) efficiently. This study provides new insights and strategies for the efficient remediation of arsenic pollution in aquatic environments.

## 1. Introduction

Arsenic (As) is a naturally occurring, highly toxic metalloid. As is categorized as a class I human carcinogen by the World Health Organization (WHO) and is a serious threat to human health. Anthropogenic activities, such as As product utilization, As-containing ore mining, and industrial As discharge, are the main sources of As pollution in the environment [1]. As exposure can lead to cancer as well as other respiratory-, digestive-, and skin-related diseases [2]. The WHO drinking water limit for As is 10 μg/L [3], but this limit is often exceeded. Currently, approximately 150 million people are exposed to As contamination globally, especially in Bangladesh, India, China, Vietnam, and the United States [4]. Therefore, it is important to develop an effective remediation technology against As pollution.

As mainly exists as arsenite ((As(III)) and arsenate (As(V)) in nature, with As(III) being much more toxic than As(V) [5,6]. As(III) can strongly interact with sulfhydryl, which leads to protein inactivation, while the structural similarities of As(V) with phosphates result in their replacement with As(V) in biomacromolecules [7,8,9,10]. The valence states of As in the environment are dependent on the redox potential and pH. Under reducing conditions, As exists in an uncharged state as As(III) (e.g., H_3_AsO_3_), while it exists in a charged state as As(V) under oxidizing conditions (H_2_AsO_4_^−^ and HAsO_4_^2−^) [11]. Therefore, As(V) can be more easily removed than As(III) [6]. At present, the As pollution of water can be remediated through coagulation precipitation, membrane separation, ion exchange, and adsorption [12,13]. Both coagulation precipitation and membrane separation are simple to perform, but the former can produce high amounts of As-containing waste, and the latter is too expensive. Ion exchange can only remediate As(V). Adsorption, namely iron-containing absorbents, is one of the most frequently used remediation methods for As pollution [14,15,16,17,18]. Iron (Fe), the fourth most abundant chemical element in the Earth’s crust, is mainly found in iron ores, such as ferrihydrite, goethite, and hematite [19]. The domain valence states of Fe in nature are ferrous iron (Fe(II)) and ferric iron (Fe(III)). Fe(II)-oxidizing bacteria can oxidize Fe(II) into Fe(III), resulting in biological Fe(III) precipitates [20]. Biological Fe(III) precipitates produced by Fe(II)-oxidizing bacteria mainly include iron oxides and large amounts of organic matter, which exhibit unique metal retention properties [21]. Compared to iron oxides generated using chemical methods, these biological iron oxides have higher specific surface areas, finer particle sizes, and higher binding energy levels [14]. Due to these properties, their heavy metal adsorption capacities have received extensive attention [20].

The formation of iron oxide precipitates by Fe(II)-oxidizing bacteria can synchronously remediate As pollution through the adsorption process. In recent years, Fe(II)-oxidizing bacteria have been studied in As remediation [22,23,24,25]. These works focused on As bioremediation using a co-culture of Fe(II)-oxidizing bacteria and As, which catalyzed biological Fe(II) oxidation to produce iron precipitates to remove As [22,23,24]. Fe(II)-oxidizing bacteria exhibit promising potential in the remediation of arsenic pollution. Thus, it is worth studying more species of Fe(II)-oxidizing bacteria. It has been reported that the iron oxide precipitates by Fe(II)-oxidizing bacteria were formed by producing extracellular enzymes [22,23,24]. However, the removal efficiency of As using extracellular enzymes to oxidize Fe(II) is unclear. 

In this study, Fe(II)-oxidizing bacteria, the strain Fe7, were isolated from iron mine soil. The strain Fe7 was identified, and its physicochemical properties were analyzed. The biological iron precipitates produced by the strain Fe7 were further analyzed through scanning electron microscopy (SEM), X-ray diffraction (XRD), X-ray photoelectron spectroscopy (XPS), and energy-dispersive X-ray spectroscopy (EDS) assays. The capacities of both Fe(II) oxidation and As removal were investigated in a co-culture of the strain Fe7 and As using iron precipitates produced by the strain Fe7 or extracellular enzymes extracted from the strain Fe7.

## 2. Materials and Methods

### 2.1. Isolation and Identification of Strain Fe7

Iron mine soil samples were collected from Ezhou City (Hubei, China; N 30°28′53″, E 114°23′3″). The pH of these samples was detected with a pH meter (METTLER TOLEDO, Columbus, OH, USA) according to the standard protocol, and it was 6.72. The soil samples were suspended in 0.9% (*w*/*v*) NaCl, and then serially diluted with 0.9% (*w*/*v*) NaCl to 10^−1^, 10^−2^, 10^−3^, 10^−4^, and 10^−5^. The 100 μL diluted samples were plated in modified Winogradsky agar medium (0.5 g L^−1^ K_2_HPO_4_, 0.5 g L^−1^ MgSO_4_·7H_2_O, 0.5 g L^−1^ NaNO_3_, 0.2 g L^−1^ CaCl_2_·2H_2_O, 0.5 g L^−1^ NH_4_NO_3_, 10 g L^−1^ ammonium ferric citrate (FeC_6_H_5_O_7_·NH_4_OH), and 15 g L^−1^ agar) [26] and incubated at 28 °C for 7 d. The pH of the medium was adjusted to 7.0 with 10 M NaOH solution. Fe(II)-oxidizing bacteria oxidized Fe(II) to Fe(III), resulting in reddish brown colonies forming in the modified Winogradsky agar medium. Thus, these reddish-brown colonies were selected for multiple separation and purification in Luria–Bertani (LB) agar medium. The LB solid medium consisted of 5 g L^−1^ yeast, 10 g L^−1^ peptone, 10 g L^−1^ NaCl, and 15 g L^−1^ agar, while the LB liquid medium did not contain agar. Finally, Fe(II)-oxidizing bacteria, the strain Fe7, were isolated and then stored with glycerol (25%, *w*/*v*) at −80 °C. All the mediums were autoclaved at 121 °C for 20 min.

The strain Fe7 was identified via sequencing its genome and analyzing its 16S rRNA gene sequences. The genomic DNA of the strain Fe7 was extracted using a QIAamp kit (Qiagen, Hilden, Germany) according to the standard protocol and sequenced using Oxford Nanopore GridION by Wuhan Bio-Broad Co., Ltd. (Wuhan, China). The nanopore sequencing library was prepared using a Ligation Sequencing kit (Oxford Nanopore Technologies, ONT; Oxford, UK, SQK-LSK110). In total, 89,347 reads with 453,718,068 bases were obtained with the GridION nanopore sequencer (ONT, Oxford, UK). To ensure the accuracy of the subsequent assembly, the raw sequencing data were filtered, and the obtained high-quality data were de novo assembled using Canu version 2.2 with default parameters [27]. The genome was annotated using the NCBI Prokaryotic Genome Annotation Pipeline in combination with GeneMarkS+ [28,29,30]. A graphical circular map of the strain Fe7 genome was generated with Proksee [31]. 16S rRNA gene sequences of the strain Fe7 were extracted from its genomic sequence. Then, a phylogenetic tree of the strain Fe7 based on its 16S rRNA gene sequences was constructed as a neighbor-joining (NJ) tree using MEGA version 11.0 software [32].

The strain Fe7 was cultivated in LB medium at 28 °C while shaking at 150 rpm. When the OD_600_ of the culture reached approximately 1.0, the culture was used to measure the morphological, physiological, and biochemical characteristics. Cells were harvested via centrifugation at 8000× *g* for 5 min at 4 °C, and the cell morphology was observed via SEM (Wuhan Detection of Technical Sousepad Ltd., Wuhan, China). The strain Fe7 was streaked on an LB plate and cultivated at 28 °C for 5 days to observe its colony morphology. Gram staining was performed using a Gram-staining kit (Jiancheng Biotech, Nanjing, China) according to the standard protocol. A motility assay was performed on 0.3% LB agar. The strain Fe7 was inoculated in LB agar and cultivated at various growth temperatures (4, 15, 20, 25, 28, 37, and 40 °C) and growth pH (4–10 at 1 pH unit increments, adjusted with 0.1 M citric acid/0.2 M Na_2_HPO_4_, pH 4.0–7.0; 0.2 M Tris/0.2 M HCl, pH 8.0–9.0; and 0.05 M NaHCO_3_/0.1 M NaOH, pH 10.0) for 7 days. Subsequently, 1% (*v*/*v*) strain Fe7 was inoculated into LB broth at different NaCl concentrations (0–1% at 0.1% concentration intervals and 1–5% at 1% intervals, *w*/*v*) and cultivated at 28 °C for 7 days. The enzyme activity and carbon-source assimilation levels were detected using an API 20NE kit (bioMérieux, Marcy-l’Étoile, France) according to the standard protocol. All the experiments were performed in triplicate.

### 2.2. Formation and Characteristics of Biological Iron Precipitates Produced by Strain Fe7

The strain Fe7 was cultivated in modified peptone yeast chromogenic medium (PCYM) to observe and identify the biological iron precipitates formed. The modified PCYM medium consisted of 0.5 g of peptone, 0.3 g of glucose, 0.2 g of yeast extract, 0.2 g of MnSO_4_·H_2_O, 0.1 g of K_2_HPO_4_, 0.2 g of MgSO_4_·7H_2_O, 0.2 g of NaNO_3_, 0.1 g of CaCl_2_, 0.1 g of (NH_4_)_2_CO_3_, and 0.8 g of ammonium ferric citrate (FeC_6_H_5_O_7_·NH_4_OH) [33]. The pH of the medium was adjusted to 7.0 with 10 M NaOH solution. The strain Fe7 was cultivated in LB medium at 28 °C with shaking at 150 rpm. When the OD_600_ of the culture reached approximately 1.0, the cells were harvested via centrifugation at 8000 g for 5 min, washed three times with 0.9% (*w*/*v*) NaCl solution, and suspended in 0.9% (*w*/*v*) NaCl solution to reach the same OD_600_. In addition, 1% (*v*/*v*) suspension was inoculated into the modified PCYM medium and cultivated at 28 °C while shaking at 150 rpm. The culture samples were collected at the indicated times, and the total iron concentration was measured using a spectrophotometer (UV1900; AOE Instruments, Shanghai, China) according to the surface water environmental quality standard (GB3838-2002) [34]. The iron precipitates produced by the strain Fe7 were harvested via centrifugation at 1000× *g* for 10 min, washed three times with ddH_2_O, and subjected to XRD, XPS, and EDS analyses conducted by Wuhan Detection of Technical Sousepad Ltd.

### 2.3. As Removal with Strain Fe7 and As Co-Culture or Using Iron Precipitates Produced by Strain Fe7

The same inoculation method used for the formation of biological iron precipitates produced by the strain Fe7 was adopted for the As removal assay. For the strain Fe7 and As co-culture experiments, the strain Fe7 and As were added to the modified PCYM medium together at the beginning. Thus, 1% (*v*/*v*) cell culture was inoculated into the modified PCYM medium containing 5 mg L^−1^ As (As(III), NaAsO_2_ or As(V), HAsNaO_4_·7H_2_O) and incubated at 28 °C while shaking at 150 rpm. The culture samples were collected at the indicated times for the measurement of the total iron and total As concentrations. To determine the As removal ability of the strain Fe7 and biological iron precipitates, the 1% (*v*/*v*) cell culture was inoculated into the modified PCYM medium and cultivated at 28 °C while shaking at 150 rpm for 72 h. The cultures were centrifuged at 1000 rpm for 10 min to collect the biological iron precipitates, and the cells were harvested via centrifugation at 8000× *g* for 5 min at 4 °C. The biological iron precipitates and cells were, respectively, added to the solutions with 5 mg/L As(III) or As(V), i.e., the As pollution systems. Both systems were cultivated at 28 °C while shaking at 150 rpm, and the samples were collected at the indicated times for the measurement of the total As and total Fe concentrations. The total As concentration was measured using high-performance liquid chromatography in combination with hydride generation atomic fluorescence spectroscopy (HPLC–HG–AFS, Beijing Titan Instruments, Beijing, China) [35]. 

### 2.4. Localization of Fe(II)-Oxidizing Enzymes in Strain Fe7

The extracellular and intracellular enzyme extracts of the strain Fe7 were collected and, respectively, added to the 271 mg L^−1^ FeSO_4_ solution. The same inoculation method used for the formation of biological iron precipitates by the strain Fe7 was adopted for the localization of the Fe(II)-oxidizing enzyme assay. First, 1% (*v*/*v*) suspension was inoculated into the modified PCYM medium and cultivated at 28 °C while shaking at 150 rpm for 18 h. The cultures were then centrifuged at 1000 rpm for 10 min to remove the iron precipitate, and the supernatant was harvested and centrifuged at 5000 rpm for 10 min. Different volumes of supernatant were added to FeSO_4_ solution as an extracellular enzyme extract to detect Fe(II) oxidation, with the FeSO_4_ solution containing no additional extracellular enzyme extract used as the control. The centrifugated cells were suspended in 0.05 M Na_2_HPO_4_/NaH_2_PO_4_ buffer solution (pH 7.0), and then cracked. The cracked cells were added to FeSO_4_ solution as an intracellular enzyme extract, while the same volume of buffer solution was added to the FeSO_4_ solution as a control. After a 1 h reaction at 30 °C, the suspension was filtered through a 0.22 μm membrane, and the Fe(II) concentration was measured with a spectrophotometer (UV1900; AOE Instruments, Shanghai, China) using the 1,10-phenanthroline spectrophotometric method according to the surface water environmental quality standard [36].

### 2.5. As Removal through the Addition of Extracellular Enzymes Produced by Strain Fe7

The extracellular enzyme extracts produced by the strain Fe7 were obtained when the strain Fe7 was cultivated in modified PCYM medium at 28 °C while shaking at 150 rpm for 24 h. Then, 2% (*v*/*v*), 5% (*v*/*v*), or 10% (*v*/*v*) extracellular enzyme extract was added to 20 mL of FeSO_4_ solution with 2 mg L^−1^ As(III) or As(V), i.e., the As(V) pollution systems. FeSO_4_ solution with 2 mg L^−1^ As(III) or As(V) free of extracellular enzyme extract was used as the control. These reaction systems were cultivated at 28 °C while shaking at 150 rpm for 3 h, and the supernatant samples were collected via centrifugation at the indicated times to measure the total Fe and total As concentrations.

For the desorption experiments on As(III) and As(V), the reaction systems with adding 5% (*v*/*v*) extracellular enzyme extract were centrifuged at 1000 rpm for 10 min at 3 h to collect As–iron precipitates. Then, these As–iron precipitates were added to 20 mL of the pH 5.0, pH 9.0 ddH_2_O (adjusted with 1 M HCl and 1 M NaOH), and 0.01 M NaH_2_PO_4_ solution, respectively. The desorption samples were reacted at 28 °C while shaking at 150 rpm for 12 h, and the supernatant were collected to measure the total As concentrations.

## 3. Results

### 3.1. Identification and Characterization of Pseudarthrobacter sp. Fe7

The genome sequences of the strain Fe7 were annotated using the NCBI Prokaryotic Genome Annotation Pipeline and deposited in the DDBJ/EMBL/GenBank under the accession numbers CP099977.1 (chromosome) and CP099978.1 (plasmid). Detailed information on the strain Fe7 genome is shown in Table 1, and graphical circular maps of the genome are provided in Figure 1A. The 16S rRNA gene sequences of the strain Fe7 were extracted from its genome, and then analyzed using the EzBioCloud and NCBI databases. The 16S rRNA gene sequences (1522 bp) of the strain Fe7 showed many similarities to *Pseudarthrobacter niigatensis* LC4 (99.03%), *Pseudarthrobacter defluvii* 4C1-a (99.03%), and *Pseudarthrobacter siccitolerans* 4J27 (98.55%). NJ phylogenetic analysis showed that strain Fe7 was closely related to *Pseudarthrobacter niigatensis* LC4 and *Pseudarthrobacter defluvii* 4C1-a (Figure 1A). Based on 16S rRNA gene sequence similarities and phylogenetic analysis, we classified the strain Fe7 into the genus *Pseudarthrobacter*.

The strain Fe7 colonies were protruding, white, and circular, with a diameter of about 1.5 mm (Appendix A). The strain Fe7 bacteria were Gram-positive, aerobic, rod-shaped (0.3–0.5 μm in diameter and 1.1–2.1 μm in length), non-flagellar, non-motile, catalase-positive, and oxidase-negative (Appendix A). They grew at 15–37 °C and pH 6.0–9.0, with optimal growth observed at 28 °C and pH 7.0. The NaCl tolerance of the strain Fe7 reached 5% (*w*/*v*). The strain Fe7 could reduce nitrate, but could not reduce nitrite or produce H_2_S. The API 20NE (bioMérieux) assay showed that the strain Fe7 was positive for esculin and β-galactosidase, but negative for indole, arginine dihydrolase, urease, and gelatin. Furthermore, this strain could use D-glucose, L-arabinose, D-mannose, D-mannitol, gluconate, maltose, malic acid, or trisodium citrate as the sole carbon source, but not N-acetyl-glucosamine, capric acid, adipic acid, or phenylacetate. The characteristic details of the strain Fe7 are presented in Table 2.

### 3.2. Formation of Biological Iron Precipitates by Strain Fe7

The strain Fe7 was cultivated in modified PCYM liquid medium to measure its ability to precipitate iron. The medium without the addition of the strain Fe7 remained clear and red (Appendix A), while the medium with the strain Fe7 added turned turbid and formed many red-brown precipitates after 48 h (Appendix A). The total iron concentration in the medium without the strain Fe7 was almost unchanged, while that in the medium with the strain Fe7 added decreased from 171 to 0.02 mg/L after 48 h, with an Fe removal rate of almost 100% (Figure 2A). These results indicate that the strain Fe7 catalyzed the formation of iron precipitates, and further SEM, XRD, XPS, and EDS analyses were performed to analyze these iron precipitates. SEM analysis showed that the iron precipitates were irregular clumps with rough surfaces and attached particles (Appendix A). XRD analysis indicated that these biological iron precipitates had a very poor crystallinity, exhibiting only one weak broad peak at 23° (Figure 2B). XPS analysis revealed the presence of C, O, N, P, and Fe in the iron precipitates produced by the strain Fe7. Furthermore, the binding energy of Fe 2p at 711 eV indicated the presence of Fe(III) in the iron precipitates produced by the strain Fe7 (Figure 2C) [37]. EDS analysis showed that the iron precipitates were mainly composed of O (37.17%), Fe (29.34%), C (16.16%), P (6.17%), and N (0.86%), suggesting that iron oxide precipitates contained a high amount of biological impurities, such as cells and proteins (Figure 2D).

### 3.3. As Removal Using Strain Fe7 and As Co-Culture

As shown in Figure 3, As(III) and As(V) could be removed using the strain Fe7 and As co-culture in the modified PYCM medium. The total Fe and total As concentrations remained unchanged in the control medium (without the strain Fe7) in both the As(III) and As(V) removal treatments (Figure 3). In the As(III) treatment, the concentration of total Fe decreased slowly during the initial 12 h after the addition of the strain Fe7, and then decreased sharply from 12 h to 48 h, accompanied by a large amount of iron precipitate formation (Figure 3A). The Fe removal rate was almost 100% after 72 h. Meanwhile, the total As concentration also decreased sharply from 12 h to 48 h, with a final As removal rate of 99.9% after 72 h (Figure 3B). In the As(V) treatment, the total Fe and total As concentrations hardly decreased in the initial 24 h (Figure 3C,D). The total Fe concentration decreased sharply from 24 h to 60 h, and a large amount of iron precipitate formed, with a final Fe removal rate of almost 98.4% after 72 h (Figure 3C), while the As(V) concentration sharply decreased with a final As removal rate of 96.7% after 72 h (Figure 3D). These results indicate the successful removal of As(III) and As(V) with the strain Fe7 and As co-culture.

### 3.4. As Removal with Iron Precipitates Produced by Strain Fe7

The strain Fe7 cells and biological iron precipitates produced by the strain Fe7 were, respectively, added to the As(III) and As(V) pollution systems. The concentration of total As in both the As pollution systems remained unchanged with the addition of the strain Fe7 cells (Figure 4A,B). When biological iron precipitates were added to the As(III) and As(V) pollution systems, the total As removal rates were 35.9% and 55.8% after 1 h, respectively, with complete removal observed after 3 h (Figure 3B). The iron precipitates in these As pollution systems were then analyzed via XPS to confirm the presence of As in these precipitates (Figure 4C,D). These results indicate that the iron precipitates produced by the strain Fe7 could efficiently remove As(III) and As(V), while the strain Fe7 cells could not remove As(III) or As(V).

### 3.5. Localization of Fe(II) Oxidizing Enzymes in Strain Fe7

The extracellular and intracellular enzyme extracts were used to investigate the Fe(II) oxidation ability of the strain Fe7. When the extracellular enzyme extract was added, the FeSO_4_ solution became turbid and gradually formed yellow iron precipitates. However, the FeSO_4_ solution without the addition of the extracellular enzyme extract (i.e., the control solution) was still clear and transparent without the formation of Fe(III) precipitates. Further analysis showed that the Fe(II) concentration in the control solution was relatively stable (100 mg/L), while the Fe(II) concentration in the FeSO_4_ solution decreased rapidly with the increasing dosage of the extracellular enzyme extract (Figure 5A). The Fe(II) concentration in the FeSO_4_ solution decreased by 28.9% when 1 mL of extracellular enzyme extract was added, and it decreased by 100% when 5 mL of extracellular enzyme extract was added. As shown in Figure 5B, the Fe(II) oxidation efficiency showed no significant difference between the phosphate buffer and control solution (without the addition of intracellular enzyme extract) with an increasing intracellular enzyme extract concentration. However, Fe precipitates were rapidly produced when 1 mL of phosphate buffer with or without intracellular enzyme extract was added to the FeSO_4_ solution with the both concentrations of Fe(II) decreasing by 98.8%. Therefore, the phosphate buffer independently promoted Fe(II) oxidation in an intracellular enzyme extract system. Thus, Fe(II) was oxidized more by the extracellular enzyme extract of the strain Fe7 rather than that of the intracellular enzyme extract.

### 3.6. As Removal through the Addition of Extracellular Enzymes Produced by Strain Fe7

As shown in Figure 5, Fe(II) was oxidized by the extracellular extract of the strain Fe7. The removal of As through the oxidation of Fe(II) by the extracellular enzyme extract of the strain Fe7 is shown in Figure 6. In the As(III) and As(V) pollution systems without the addition of the extracellular enzyme extract, the concentrations of total Fe and total As remained relatively unchanged. In the As(III) pollution system with 2% (*v*/*v*) extracellular enzyme extract, the total Fe and As removal rates were, respectively, 76.9% and 98.6% after 2 h, which increased to 92.6% and 100% with the 5% (*v*/*v*) extracellular enzyme extract addition and increased further to 99.6% and 100% with the 10% (*v*/*v*) extracellular enzyme extract addition (Figure 6A,C). In the As(V) pollution system with 2% (*v*/*v*) extracellular enzyme extract, the total Fe and As removal rates were, respectively, 78.4% and 99.4% after 2 h, which increased to 93.0% and 100% with the 5% (*v*/*v*) extracellular enzyme extract addition and increased further to 99.4% and 100% with the 10% (*v*/*v*) extracellular enzyme extract addition (Figure 6B,D). These results indicated that the extracellular enzyme extracts produced by the strain Fe7 could effectively remediate the As pollution of water.

The aforementioned iron precipitates that absorbed As were subsequently added to the solutions with a pH 5.0, pH 9.0, or 0.01 M NaH_2_PO_4_ solution. The desorption effects of As(III) and As(V) in the different pH solutions and NaH_2_PO_4_ solutions are shown as follows: The desorption rates of As(III) in the pH 5.0, pH 9.0, and NaH_2_PO_4_ solutions were 19.7%, 22.8%, and 56.1%, respectively. The desorption rates of As(V) in the pH 5.0, pH 9.0, and NaH_2_PO_4_ solutions were 0%, 65.9%, and 21.8%, respectively. Thus, the NaH_2_PO_4_ solution exhibited a superior desorption ability for As(III), whereas the pH 9.0 solution had an optimal desorption capability for As(V).

## 4. Discussion

*Pseudarthrobacter* sp. Fe7 oxidized Fe(II) to produce biological Fe(III) precipitates (Figure 2 and Appendix A). The oxidizing of Fe(II) using the enzymes in the strain Fe7 occurred in the extracellular extracts rather than the intracellular extracts (Figure 5). Our results are similar to the previous reports on Fe(II) oxidation by other bacteria, such as some of *Pseudomonas* species [24] and *Sphaerotilus natans* Z1 [23]. The study of these extracellular extracts showed that the phosphate buffer promoted Fe(II) oxidation (Figure 5B). The addition of phosphate buffer can release OH^−^ from an FeSO_4_ solution, promoting the formation of Fe(OH)_2_, which can be oxidized by O_2_ to form Fe(III) precipitates. Since the biological Fe(III) oxides are usually insoluble at a circumneutral pH [38], Fe(II) oxidation occurred extracellularly in this study, thereby avoiding the accumulation of Fe(III) precipitates to hinder cell metabolism.

Fe(II)-oxidizing bacteria, such as *Sphaerotilus natans* Z1, some of the *Pseudomonas* species, and the *Pseudomonas* sp. strain GE-1, can oxidize Fe(II) to produce iron precipitates [23,24,39]. The iron precipitates produced by these strains exhibited two XRD peaks at 35° and 62° and were identified as ferrihydrite [40]. In our study, the iron precipitates produced by the strain Fe7 showed only one weak broad peak at 23°, which did not correspond with ferrihydrite (Figure 2B). Comparing the characteristic XRD peaks of the different iron minerals, such as hematite [37], magnetite [37], goethite [41], and lepidocrocite [42], the iron precipitates produced by the strain Fe7 did not correspond to these iron minerals. Previously studies showed that the main products of the biological oxidation of iron are usually a mixture of poorly ordered iron oxides and often contain significant amounts of organic matter. Our EDS analysis revealed that the iron precipitates produced by the strain Fe7 contained biological impurities, which led to its poor crystallinity. The results are similar to the iron precipitates produced by *Sphaerotilus natans* Z1, some of the *Pseudomonas* species., and the *Pseudomonas* sp. strain GE-1 [23,24,39]. The iron precipitates produced by the strain Fe7 were an unknown reddish-brown indicative of the Fe(III) mineral and had poor crystallinity.

For the Fe(II)-oxidizing bacteria, there are two main mechanisms for the removal of As using the bacteria–As co-culture: the bacterial cells absorb As, or the iron precipitates produced by these bacteria absorb and precipitate As. Figure 3 showed that the total As concentration decreased with the formation of iron precipitates by the strain Fe7 and As co-culture. As was removed by the iron precipitates produced by the strain Fe7 rather than by the cells of the strain Fe7 in the As pollution systems (Figure 4). The iron precipitates produced by the strain Fe7 could absorb and co-precipitate to remove As. The cells could not absorb As or produced enzymes to form iron precipitate under non-culture conditions, leading to the low removal rates of As in the As pollution system. Thus, the removal of As with the strain Fe7 and As co-culture was performed with the iron precipitates produced by the strain Fe7. The results were consistent with that of the strain GE-1 [39], but differed from that of *Sphaerotilus natans* Z1 [23]. Both the cells of the strain Z1 and the iron precipitates produced by the strain Z1 could absorb, and thus, remove As [23].

At present, there are some reports about the use of Fe(II)-oxidizing bacteria to remediate As pollution. For example, *Gallionella ferruginea* and *Leptothrix ochracea* can be used as filtration media to remove 95.0% of the As (As concentration is 50–200 µg/L) in water [43,44]. Under the condition of using a bacteria-As co-culture, strain GE-1 could remove As by producing ferrihydrite, and the removal efficiency of As could reach 100% after 96 h (As concentration is 1 mg/L) [39]. In our study, when the strain Fe7 and As were co-cultured, the As could not be completely removed after 72 h. However, the iron precipitates collected when the strain Fe7 was cultured for 72 h could completely remove the As after 3 h. These results demonstrate the superior As removal ability of the iron precipitates produced by the strain Fe7 compared to that of the bacteria–As co-culture. In addition, different volumes of extracellular enzyme extract were added to the As solution systems. When 2% (*v*/*v*) extracellular enzyme extract was added to the As(III) and As(V) pollution systems, 76.9 and 78.4 mg/L of Fe were, respectively, precipitated. The corresponding residual amounts of As were 28.0 and 11.2 μg/L, which do not meet the WHO drinking water standard [3]. When 5% (*v*/*v*) or 10% (*v*/*v*) extracellular enzyme extracts were added to the As(III) and As(V) pollution system, the As could be removed completely. Thus, 5% (*v*/*v*) extracellular enzyme extract was selected to remediate the As pollution of water.

Iron-containing materials exhibit excellent properties in removing As. Due to the complexity of the environment, the adsorbed As may be released into the environment via desorption under certain conditions. We investigated the desorption abilities of the iron precipitates produced by adding 5% (*v*/*v*) extracellular enzyme extract in our research. The results demonstrated that both As(III) and As(V) could be desorbed in 0.01 M NaH_2_PO_4_ solution. Since As and phosphorus (P) belong to the main group VA in the periodic table of elements, PO_4_^3−^ is a strong ligand that competes with As for adsorption sites because of its similar outer electronic structure [45,46]. Additionally, we found that As(III) could be desorbed in the pH 5.0, pH 9.0, and 0.01 M NaH_2_PO_4_ solutions, with the NaH_2_PO_4_ solution exhibiting the strongest desorption ability among them all. However, As(V) could not be desorbed in a pH 5.0 solution, and the strongest desorption ability for As(V) was observed in a pH 9.0 solution. The different desorbed behavior observed between As(III) and As(V) during our study is consistent with the previous findings on various iron minerals, such as limonite, siderite, hematite, and magnetite [47]. Furthermore, we discovered that both As(III) and As(V) could be desorbed from several iron minerals in the pH 5.0, pH 9.0, and 0.01 M NaH_2_PO_4_ solutions. However, the As(V) could not be desorbed from the iron precipitates produced by an extracellular enzyme extract of the strain Fe7 in a pH 5.0 solution. Thus, the iron precipitates produced by the extracellular enzyme extract of the strain Fe7 were more suitable for the remediating of As(V) pollution water, particularly under acidic conditions.

## 5. Conclusions

The strain Fe7, an Fe(II)-oxidizing bacteria belonging to the genus *Pseudarthrobacter*, was isolated from iron mine soil and identified as a Gram-positive, rod-shaped, aerobic bacterium. The extracellular enzyme extracts of the strain Fe7 oxidized Fe(II) into Fe(III) to produce iron oxide precipitates with poor crystallinity. Different methods were explored in the removal of As(III) and As(V) using the strain Fe7: (1) a strain Fe7 and As co-culture, (2) iron oxide precipitates produced by the strain Fe7, and (3) iron oxide precipitates produced by the extracellular enzyme extract of the strain Fe7. The investigation of *Pseudarthrobacter* sp. Fe7 provides new material and selectable remediation strategies for the effective bioremediation of the As pollution of water.

## Figures and Tables

**Figure 1 microorganisms-11-02860-f001:**
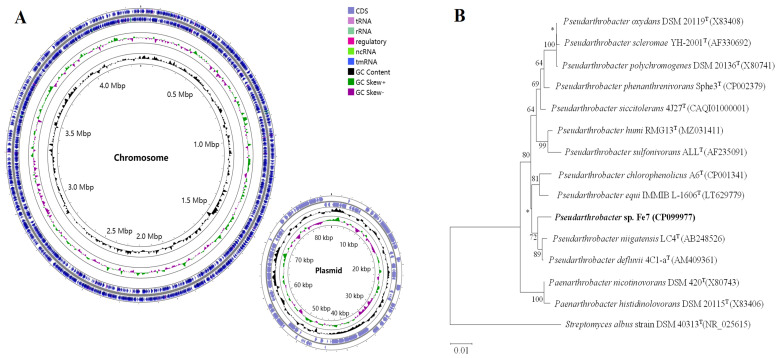
Graphical circular map of the genomic sequence of strain Fe7 and NJ phylogenetic tree of strain Fe7. (**A**) Graphical circular map of the genomic sequence of strain Fe7 (chromosome and plasmid). (**B**) NJ phylogenetic tree of strain Fe7 and related strains based on 16S rRNA gene sequences. * indicates that the bootstrap value is less than 50%. The scale bar indicates 0.01 substitution per nucleotide position.

**Figure 2 microorganisms-11-02860-f002:**
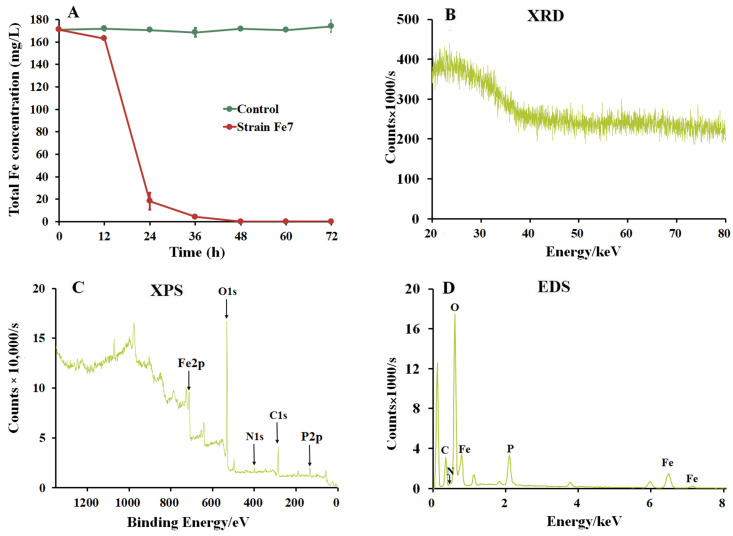
Change curve of total iron and characterization of biological iron precipitates produced by strain Fe7. (**A**) Change curves of total iron; (**B**) XRD, (**C**) XPS, and (**D**) EDS patterns.

**Figure 3 microorganisms-11-02860-f003:**
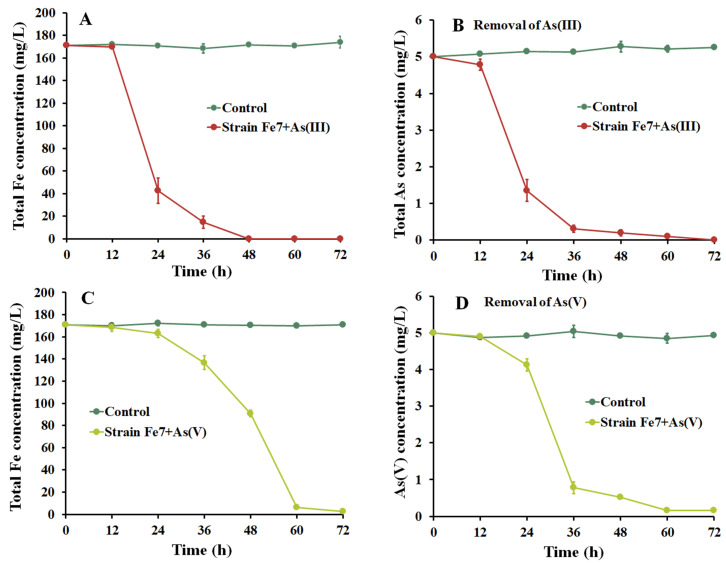
Curves of total iron and total arsenic removal. (**A**) Iron removal curves in the presence of As(III). (**B**) As(III) removal curves. (**C**) Iron removal curves in the presence of As(V). (**D**) As(V) removal curves. The data are expressed as the mean ± standard deviation (*n* = 3).

**Figure 4 microorganisms-11-02860-f004:**
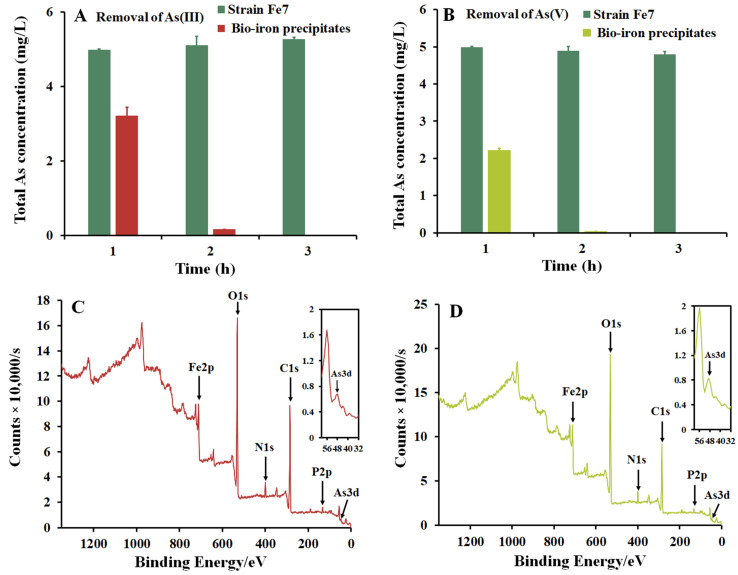
Arsenic removal by strain Fe7 cells, iron precipitates produced by strain Fe7, and the XPS patterns of the pollution systems after the addition of iron precipitates produced by strain Fe7. Total As removal in the (**A**) As(III) and (**B**) As(V) pollution systems. XPS patterns in the presence of (**C**) As(III) and (**D**) As(V). The data are expressed as the mean ± standard deviation (*n* = 3).

**Figure 5 microorganisms-11-02860-f005:**
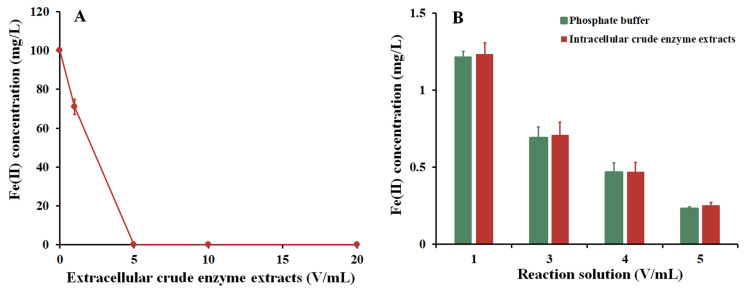
Fe(II) oxidation by the (**A**) extracellular and (**B**) intracellular enzyme extracts of strain Fe7. Data are expressed as the mean ± standard deviation (*n* = 3).

**Figure 6 microorganisms-11-02860-f006:**
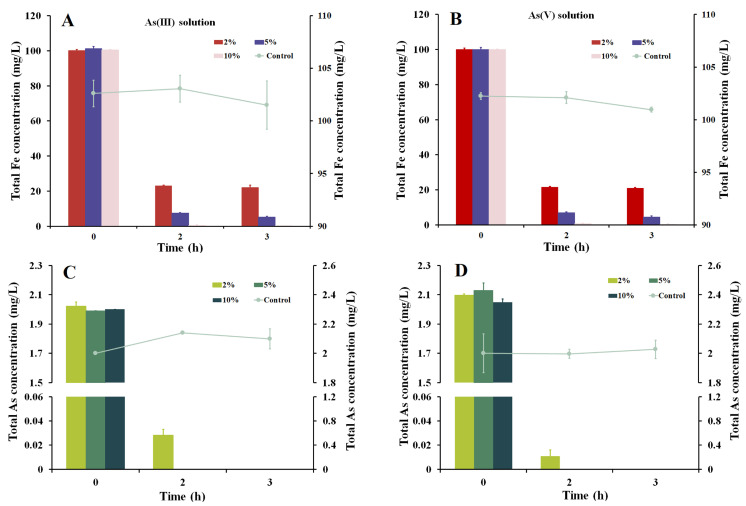
Total iron and total arsenic removal rates after the addition of the extracellular enzyme extracts of strain Fe7. Total removal rate of iron in the (**A**) As(III) and (**B**) As(V) pollution systems. Total removal rate of As in the (**C**) As(III) and (**D**) As(V) pollution systems. Data are expressed as the mean ± standard deviation (*n* = 3).

**Table 1 microorganisms-11-02860-t001:** Detailed information on the strain Fe7 genome.

Type	Size (Mb)	GC%	Protein	rRNA	tRNA	Other RNA	Gene	Pseudogene	Accession Number
Chromosome	4.38	65.8	2966	15	53	3	4292	1255	CP099977.1
Plasmid	0.09	61.8	84	-	-	-	94	10	CP099978.1

**Table 2 microorganisms-11-02860-t002:** The characteristics of the strain Fe7.

Characteristic	Strain Fe7
Gram’s reaction	+
Temp. range (°C)	15–37
pH range	6.0–9.0
NaCl range (%, *w*/*v*)	0–5
Motility	−
Nitrate reduction	+
Nitrite reduction	−
Indol production	−
H_2_S production	−
Hydrolysis of:	
Gelatin	−
Esculin	+
Enzyme activity of:	
Oxidase	−
Urease	−
β-galactosidase	+
Arginine dihydrolase	−
Assimilation of:	
Glucose	+
Arabinose	+
Mannose	+
Mannitol	+
N-acetylglucosamine	−
Maltose	+
Gluconate	+
Caprate	−
Adipic acid	−
Malate	+
Citrate	+
Phenylacetate	−

+, positive; −, negative.

## Data Availability

Data are contained within the article and Appendix A.

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
