# Peer review of "Arsenic Removal via the Biomineralization of Iron-Oxidizing Bacteria Pseudarthrobacter sp. Fe7"

_microorganisms, 2023, doi:10.3390/microorganisms11122860_

Round 1

Reviewer 1 Report

Comments and Suggestions for Authors

Paper entitled “Arsenic removal by the biomineralization of iron-oxidizing bacteria Pseudarthrobacter sp. Fe7” meets the necessary standards for publication in this journal.

  I recommend to complete the material with information from the following articles:  doi:10.3390/ma15155366 doi:10.3390/ma14133731  Figure 1 is not clear. The text can be written in larger letters. At what pH were the studies performed? Which is the removal arsenic species?  I recommend establishing the mechanism of the arsenic removal process.  Can adsorption-desorption studies be carried out?  I recommend comparison with other bacteria that can be found in the specialized literature and were used for the arsenic removal. Attention when writing references. They are not unitary.  

Comments on the Quality of English Language

 Please check the entire manuscript carefully for eventual typographical errors.

Reviewer 2 Report

Comments and Suggestions for Authors

The manuscript entitled «Arsenic removal by the biomineralization of iron-oxidizing bacteria Pseudarthrobacter sp. Fe7” is aims to As removal using Fe(II)-oxidizing bacteria Pseudarthrobacter strain Fe7 isolated from iron mine soil. The authors used a number of modern and precise methods to prove the arsenic and iron phases in biogenic sediments. This study provides new insights and strategies for the efficient remediation of arsenic pollution in  aquatic environments. To my mind this manuscript is topical and corresponding to the aims and scopes of the Microorganisms journal.

Here are the comments I found while reading the manuscript

1.     It is worth clearly writing the purpose of the work, including isolation and identification of the strain, isolation of enzymes, etc.

2.     It is necessary to describe in more detail the sampling site and physicochemical conditions. Temperature, pH, concentration of arsenic, iron and other components.

3.     It is worth linking the strain cultivation conditions with the conditions at the sampling site. For example, why was the temperature chosen to be 28, was the temperature range checked?

4.     It is worth speculating about the forms of iron in biogenic sediments. The authors noted the important role of the phosphate buffer during iron precipitation. Sources of phosphorus in mineral or organic form added to the medium may also take part in this process. The low degree of arsenic removal in the system with cells can be explained by the low concentration of phosphate and removed iron. You need to pay attention to this.

5.     Pseudomonas is best known for its ability to reduce iron under anaerobic conditions.

6.     It is worth describing in more detail the experiment with co-culture in the methods, what do you mean? Were more organisms added?

7.     Have you done experiments in hermetically sealed vials with longer cultivation? I think it would be interesting to discuss the direct enzymatic reduction of arsenate.

8.     In conclusion, it is worth briefly describing the technological prospects for using the strain for wastewater treatment.

Comments on the Quality of English Language

Minor editing of English language required

Round 2

Reviewer 2 Report

Comments and Suggestions for Authors

The authors took my comments into account and in this form I can recommend the manuscript for publication

Comments on the Quality of English Language

Minor editing of English language required